# Realization and topological properties of third-order exceptional lines embedded in exceptional surfaces

Weiyuan Tang[1,2], Kun Ding [3] ✉ & Guancong Ma [1] ✉

As the counterpart of Hermitian nodal structures, the geometry formed by exceptional points (EPs), such as exceptional lines (ELs), entails intriguing spectral topology. We report the experimental realization of order-3 exceptional lines (EL3) that are entirely embedded in order-2 exceptional surfaces (ES2) in a three-dimensional periodic synthetic momentum space. The EL3 and the concomitant ES2, together with the topology of the underlying space, prohibit the evaluation of their topology in the eigenvalue manifold by prevailing topological characterization methods. We use a winding number associated with the resultants of the Hamiltonian. This resultant winding number can be chosen to detect only the EL3 but ignores the ES2, allowing the diagnosis of the topological currents carried by the EL3, which enables the prediction of their evolution under perturbations. We further reveal the connection between the intersection multiplicity of the resultants and the winding of the resultant field around the EPs and generalize the approach for detecting and topologically characterizing higher-order EPs. Our work exemplifies the unprecedented topology of higher-order exceptional geometries and may inspire new non-Hermitian topological applications.

Spectral degeneracies in band structure often possess intriguing topological properties. For example, in Hermitian three-dimensional (3D) systems, point degeneracies such as Dirac, Weyl, or triple points are monopoles of Berry flux[1,2]. Degeneracies can form continuous geometries, e.g., nodal lines with intricate structures such as rings, links, and chains[3]. Nodal surfaces have also been shown to carry topological charges[4,5]. Recently, physicists found that non-Hermiticity further enriches the diversity of band topology[6–10]. This is partly due to the fact that the non-Hermitian spectrum occupies the complex plane, such that the energies themselves can exhibit topological winding behaviors, leading to an additional layer of "spectral topology" underneath the wavefunction topology, giving rise to skin effects[11–19] and spectral knots[20,21]. Non-Hermitian degeneracies known as EPs possess topological properties characterizable by spectral winding numbers[6,8,9,22–24]. Most studies focus on EPs formed by two coalescing

states with one being defective, i.e., defective order-2 EPs. Akin to Hermitian degeneracies, they can also form nodal structures, such as rings[25–29], lines[30–32], links and chains[30,32,33], and surfaces[22,34,35]. Higher-order EP is formed when three or more states coalesce, with two or more states being defective. Although their realizations were reported in several experiments[36–39], their stable existence demands more degrees of freedom in the parameter space or a higher level of symmetries[40–43].

Here, we report the experimental realization of EL3 entirely embedded on ES2. The EL3 (ES2) is formed by order-3 (order-2) defective EPs. Both the EL3 and ES2 run continuously through the entire 3D parameter space, which is homeomorphic to a 3-torus by design. Such geometry presents an unexpected difficulty for topological characterization. The prevailing methods that extract topological properties of nodal degeneracies are based on the principle of

[1]Department of Physics, Hong Kong Baptist University, Kowloon Tong, Hong Kong, China. [2]Department of Physics, The University of Hong Kong, Pokfulam Road, Hong Kong, China. [3]Department of Physics, State Key Laboratory of Surface Physics, and Key Laboratory of Micro and Nano Photonic Structures (Ministry of Education), Fudan University, Shanghai 200438, China. ✉e-mail: kunding@fudan.edu.cn; phgcma@hkbu.edu.hk

homotopy group. Under such an approach, topological invariants are evaluated either on a 2-sphere enclosing the entire nodal structure, with the topological charges of a Weyl point being an important case; or on a 1-sphere encircling the nodal structure, such as the characterization of topological nodal lines[3,5–7,44] and order-2 EP lines[24,29]. However, any enclosing sphere of a single EL3 would encounter ill-behaved spectral singularity on the ES2, thus defying the continuous requirement for spectral winding. In a recent study, it is theoretically shown that the resultants of the Hamiltonian matrix can be viewed as auxiliary manifolds associated with but different from the eigenvalue manifolds[40]. Here, based on the intersection multiplicity of the resultants determining the location of EPs, we further uncover that a "resultant vector field" can be uniquely chosen to vanish only at the EL3 and remain continuous at the ES2, leading to a resultant winding number for diagnosing the topology of the EL3 while ignoring the influence of the ES2. The validity of our approach is verified by successfully predicting the local evolution of a touching point (TP) of two EL3 under perturbation. Our study expands the understanding of non-Hermitian topology by unveiling novel topological scenarios exclusive to higher-order EP structures.

## Results

### Realization of symmetry-protected EL3

First, we present an experiment-feasible lattice system that realizes the EL3. We begin with a codimension analysis of $n$-fold non-Hermitian degeneracy point, denoted EP$n$. An isolated EP$n$ is found when $n$ complex eigenvalues become identical, i.e., an EP2 emerges at $\omega_1 = \omega_2$,

and an EP3 occurs with $\omega_1 = \omega_2$ and $\omega_2 = \omega_3$. In other words, an isolated EP$n$ is a common solution of a set of $n − 1$ equations to be satisfied[42,43], and the existence of such a solution requires degrees of freedom (DOFs). In the absence of any symmetry, a $d$-dimensional structure constituted by EP$n$ lives in the parameter space with minimal $2(n − 1) + d$ dimensions. Hence both isolated EP3 and ES2 are stable in a four-dimensional (4D) parameter space[45,46]. The dimensionality requirement can be reduced by enforcing additional symmetries. In particular, when parity-time symmetry is respected, the characteristic polynomial of a Hamiltonian $H$, denoted $p(\omega) = \det(H − \omega \mathbf{I}) = 0$ where $\omega$ denotes the eigenvalues and $\mathbf{I}$ is an identity matrix, has entirely real discriminant $\mathscr{D} = \prod_{\mu < \nu}(\omega_\mu − \omega_\nu)^2$ (where $\mu$ and $\nu$ are the eigenvalue indices), i.e., Im $\mathscr{D} = 0$ is always satisfied. Hence the DOF requirement is reduced to $n − 1$. Consequently, both ES2[22] and EL3 are accessible in a 3D PT-symmetric three-state system, serving as our starting point in designing an experiment-feasible lattice model.

We base our experimental system on coupled acoustic cavities[46,47]. Here, we engineer the system such that its parameter space is mapped to a 3D lattice model. We begin with three air-filled cylindrical cavities stacked together (Fig. 1a). Within each cavity, a thin plate is fixed in the radial direction to form standing-wave modes. We use the second azimuthal mode, whose velocity $v$ profiles and pressure $P$ are shown in Fig. 1b–d, respectively. The mode is harmonic in the azimuthal angle $\phi$ (Fig. 1e, f). Such mode profiles can naturally realize $2\pi$-periodic synthetic coordinates, denoted $(\phi_1, \phi_2, \phi_3)$. Because the parameter space is clearly a homeomorphism of a 3-torus, we call it a 3D synthetic Brillouin zone (SBZ) henceforth.

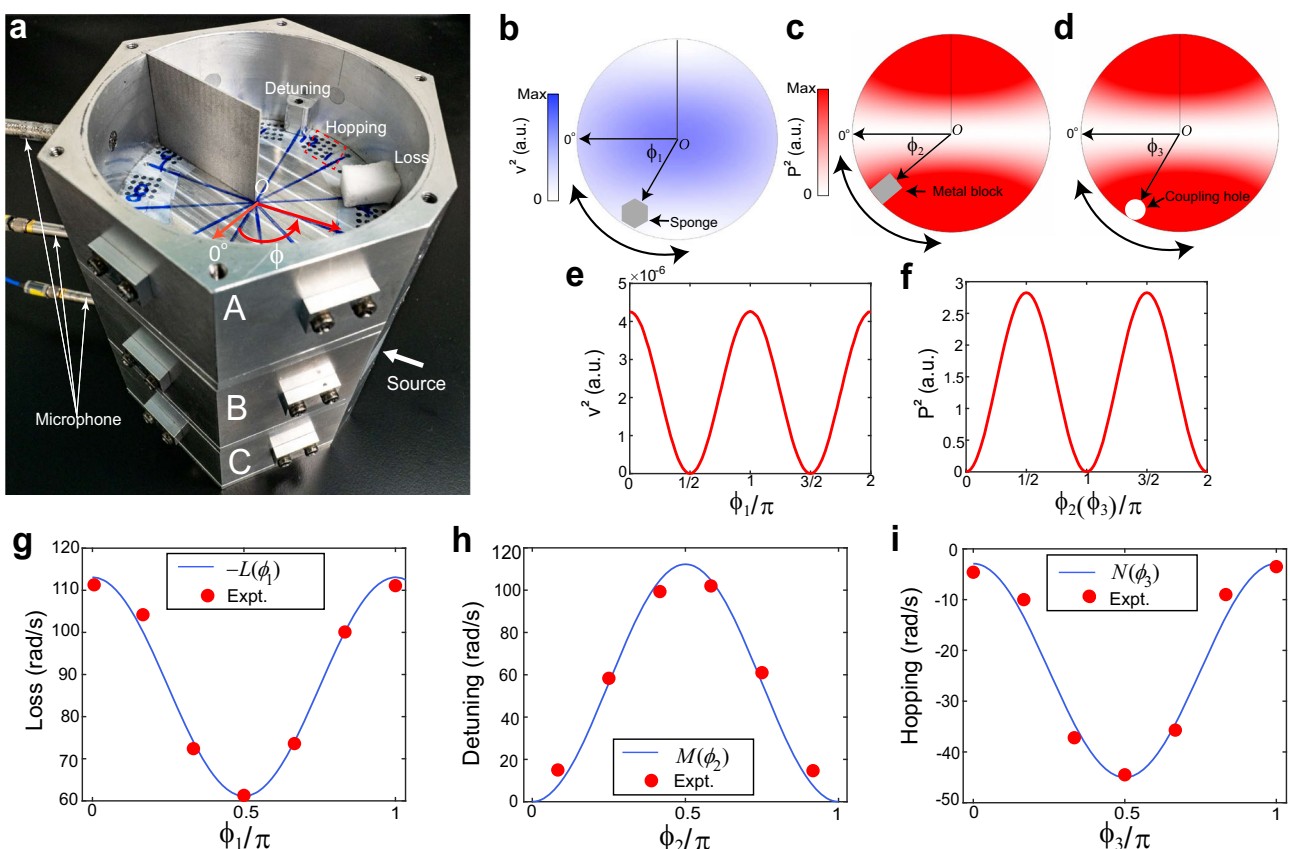

**Fig. 1 | Realization of the three-state non-Hermitian model using acoustic cavities.** A photographic image of the ternary acoustic cavity system is shown in **a**. **b** shows the cross-sectional distribution of the squared velocity of a single cavity. The azimuthal position of the sponge, which maps to $\phi_1$, tunes the dissipative rate. **c**, **d** show the squared pressure field in a single cavity. The azimuthal position of the

metal block (coupling hole), which corresponds to $\phi_2$ ($\phi_3$), tunes the resonant frequency (hopping strength). **e**, **f** respectively shows $v^2$ as a function of $\phi_1$ and $P^2$ as a function of $\phi_{2,3}$. **g–i** plot the onsite loss (**g**), onsite detuning (**h**), and hopping strength (**i**) as functions of $\phi_1$, $\phi_2$, and $\phi_3$, respectively. The blue curves are fitted from experimental data (red circles).

**Table 1 | Experimental implementations of the synthetic coordinates**

| Synthetic coordinate | Physical quantity | Physical mechanism | Function relation | Implementation |
|---|---|---|---|---|
| $\phi_1$ | Onsite dissipative rate (non-Hermiticity) | Dissipation proportional to local kinetic energy $v^2(\phi)$. | $v^2(\phi_1) \propto \cos^2(\phi_1)$ | Variation of the azimuthal position $\phi_1$ of the sponge. |
| $\phi_2$ | Onsite resonant frequency | Resonant frequency sensitive to the local pressure intensity $P^2(\phi)$. | $P^2(\phi_2) \propto \sin^2(\phi_2)$ | Variation of the azimuthal position $\phi_2$ of the metal block. |
| $\phi_3$ | Hopping | Coupling strength sensitive to the local pressure intensity $P^2(\phi)$. | $P^2(\phi_3) \propto \sin^2(\phi_3)$ | Variation of the azimuthal position $\phi_3$ of the coupling holes. |

The dissipative rate, resonant frequency, and coupling coefficients can be tuned as independent functions of $\phi_1, \phi_2, \phi_3$, respectively. Table 1 summarizes the dependence of each physical quantity on the respective synthetic coordinate and the experimental implementations. Let $\phi_1$ tune the imaginary part of onsite resonant frequency in cavities A and C, which is the source of non-Hermiticity in our system. This is achieved by placing a piece of acoustic sponge to generate losses, which are linearly proportional to the local kinetic energy[48], given as $K \propto v^2(\phi_1) \propto \cos^2\phi_1$. Let $\phi_2$ modulate the real part of the onsite resonant frequency of cavity B. A small metallic block is placed on the circumference for this purpose, and its azimuthal position is assigned as $\phi_2$. Its perturbation to the resonant frequency is linear to the local acoustic potential energy[48] $U \propto P^2(\phi_2) \propto \sin^2\phi_2$. Assign $\phi_3$ to control the coupling strength between cavities A and B, also B and C. Acoustic coupling strength is proportional to pressure intensity, i.e., $P^2(\phi_3) \propto \sin^2\phi_3$.

Our discussion is based on PT-symmetric systems. To satisfy PT symmetry, an equal amount of acoustic sponge is inserted into all three cavities as the biased loss, and then a specific amount of sponge in cavity A is relocated to C, such that an effective gain is created in A and the same amount of loss is added to C. We characterize the detuning, loss, and coupling, as well as realize the synthetic coordinate $\phi_1, \phi_2, \phi_3$ in our systems, and the results are shown in Fig. 1g–i as functions of the corresponding synthetic momenta. By tuning the acoustic parameters, the loss in the system follows $L(\phi_1) = -60.68(0.50\sin^2\phi_1 - 1)$, the detuning of cavity B is described by $M(\phi_2) = -38.62\cos^2\phi_2$, and the coupling between neighboring cavities obeys $N(\phi_3) = -42.91(1 - 0.62\cos^2\phi_3)$.

A three-state Hamiltonian $H = (\omega_0 - i\gamma_0)\mathbf{I} + H_{3b}$ captures the acoustic cavity system, where $\omega_0$ is the resonance frequency of the second azimuthal cavity mode and $\gamma_0$ is the dissipation rate. The second term is

$$H_{3b}(\phi_1, \phi_2, \phi_3) = \begin{pmatrix} iL(\phi_1) & N(\phi_3) & 0 \\ N(\phi_3) & M(\phi_2) & N(\phi_3) \\ 0 & N(\phi_3) & -iL(\phi_1) \end{pmatrix}. \qquad (1)$$

By using the trigonometric identity $\cos(2\phi) = 2\cos^2\phi - 1 = 1 - 2\sin^2\phi$, we obtain $L(\phi_1) = \gamma + 2\kappa_1\cos(2\phi_1)$, $M(\phi_2) = \epsilon + 2\kappa_2\cos(2\phi_2)$, and $N(\phi_3) = \beta + 2\kappa_3\cos(2\phi_3)$. It then follows that model (1) maps to a periodic lattice shown in Fig. 2a. Herein, the constant parameters are the onsite gain (loss) rate of site-A (C) $\gamma = 45.51$, the onsite offset to site-B $\epsilon = -19.31$, $\beta = -29.60$, $\kappa_1 = 7.59$, $\kappa_2 = -9.66$, and $\kappa_3 = 6.65$, all have the unit of rad/s and are obtained by benchmarking the experimental system. The non-Hermiticity in the model comes from the function $\pm iL(\phi_1)$, which manifests as the constant gain (loss) $\pm i\gamma$ and the asymmetric long-range hopping $\pm i2\kappa_1\cos(2\phi_1)$.

The emergence of an EPn can be identified by the conditions $\mathcal{R}[p^{(j)}, p^{(j+1)}] = 0$ with $0 \leq j < n - 1$, and $\mathcal{R}$ denotes the resultant, $p^{(j)}$ is the $j$th-order derivative of the characteristic polynomial with respect to $\omega$. As such, we identify both ES2 and EL3 in the SBZ (Fig. 2b). We note that the SBZ here is an extended BZ consisting of eight identical copies of the first BZ. Physically, this is due to the quadratic

dependence of the physical quantities (loss, hopping, and detuning) on the synthetic dimension; and mathematically, the trigonometric double-angle formulas play a role in Eq. (1). The choice of SBZ does not affect the validity of our analysis that follows. The real-eigenfrequency Riemann surfaces on three distinct 2D slices are displayed in Fig. 2c–e. Panels 2d, e show $\phi_1\phi_2$-planes sliced at $\phi_3 = \pi/2$ and $\phi_3 = \pi/3$, respectively. The remaining state (shown in orange) touches the EL2 at particular isolated points and forms EP3 (purple hexagons and red stars). These EP3 only appear when $\phi_2 = \pm\pi/2$. The conditions for EP3 to appear are $M(\phi_2) = 0$ and $L(\phi_1) \pm \sqrt{2}N(\phi_3) = 0$, where the $\pm$ sign suggests two possible solutions (Supplemental Materials). The EL3 are plotted in Fig. 2b, c. Two EL3 form a linear crossing at $\phi_1 = 0, \pm\pi$, which we denote as the TP. The TPs are previously defined to the points where two nodal lines touch[49], and here we generalize it to EL3.

The ES2 and EL3 are observed in our acoustic experiments. The acoustic pressure responses at each cavity are measured near $\omega_0$ at different synthetic momenta. The real and imaginary parts of the eigenfrequencies are then extracted from the acoustic responses using the Green's function[36,46,47] (Supplemental Materials). We fix $\phi_2 = 0.5\pi$, then choose five different $\phi_3$ indicated by the horizontal dashed lines in Fig. 3a, and for each $\phi_3$, the acoustic system is tuned to five different $\phi_1$. Both the real and imaginary parts of the eigenfrequencies from the measured data are depicted in Fig. 3b, which show good agreement with the theoretical results (solid curves). Therein, the EP3s are marked by red arrows. These positions are then marked in Fig. 3a with the stars and fall on the computed EL3. We then observe the ES2 by performing similar experiments in different $\phi_1\phi_2$-planes at $\phi_3 = 0.5\pi$ (Fig. 3c) and $\phi_3 = 0.33\pi$ (Fig. 3e), which intersect with the ES2 and the EL3. The coalescence of two of the three states or all three states is clearly seen (Fig. 3d, f), and the measured locations of the EP2s and EP3s also conform well with the theoretical results.

## Characterization of the EL3

The presence of both ES2 and EL3 gives rise to intriguing topological characteristics. The ES2 form close, continuous 2D surfaces that kiss at the TPs, which separate the eigenvalue manifold into disjoint regions. The topological properties of ES2 protected by PT symmetry are characterized by a $\mathbb{Z}_2$ topological invariant[22,35], which is equal to 1 here. Yet the topological characterization of EL3 is more challenging. The EL3 are entirely embedded in the ES2 and also osculate at the TPs. Also, both the ES2 and EL3 run through the SBZ in the $\phi_1$-direction. Such geometry entails difficulties in their topological characterization. As mentioned before, the topology of a nodal structure is diagnosed by invariants computed on the $m$-spheres with $0 \leq m \leq d - 1$, which enclose the nodal structures. Examining the ES2 and EL3, it is clear that no 2-sphere can enclose them. Yet it remains possible to encircle both the ES2 and EL3 together using the zeroth or first homotopy group (Supplemental Materials). Under the zeroth homotopy group count equivalence classes of 0-sphere, i.e., two separate points, the ES2 and EL3 together form a manifold that is $\mathbb{Z}_2$ classified[22,35]. Under the first homotopy group, a 1-sphere, i.e., a closed loop, can encircle the ES2 and EL3 together. An example is shown in Fig. 2b as the green dashed loop. We have computed the eigenvalue winding number, defined as

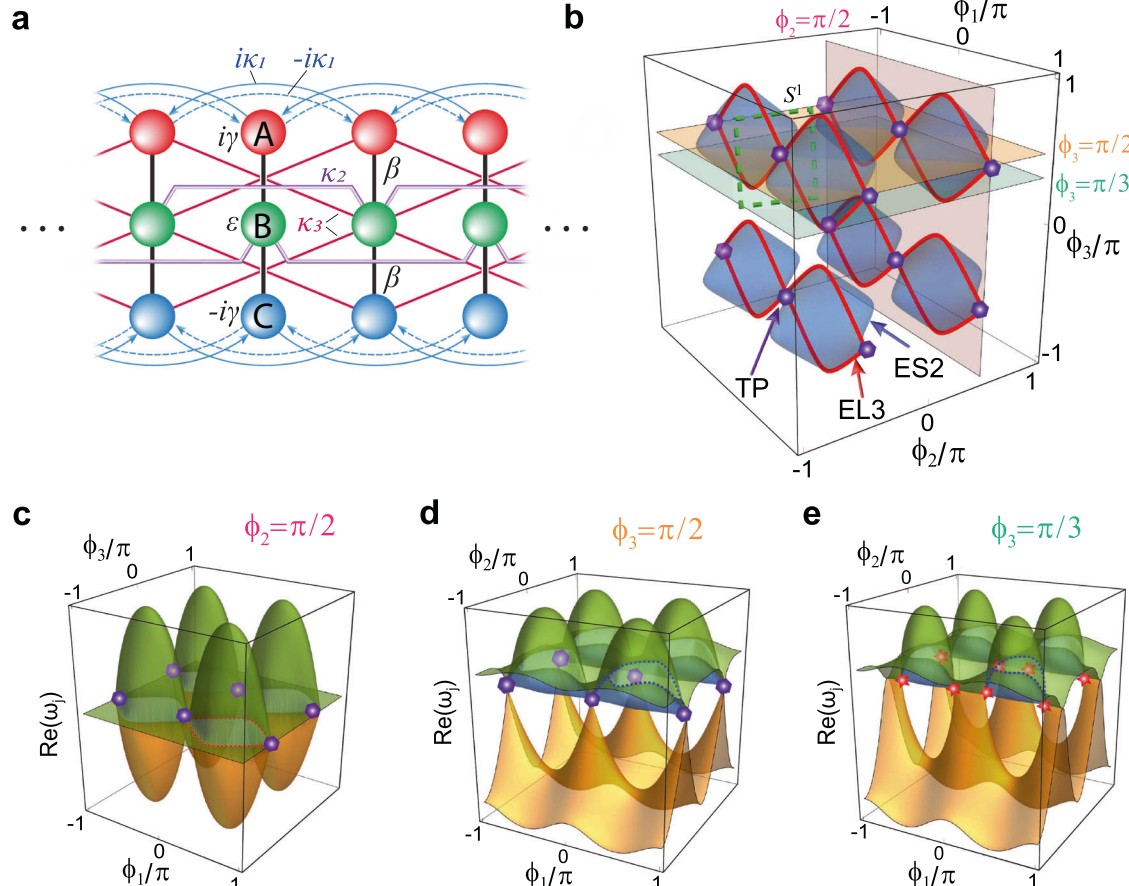

**Fig. 2 | EL3 and ES2 in the SBZ mapped to a 3D lattice model. a** The lattice that maps to Eq. (1). **b** The EL3 (red curves) and ES2 (blue surfaces) in the SBZ. The purple hexagons denote the TPs of EL3. The green dashed line denotes an $S^1$ loop encircling a TP. There are eight identical copies of EL3 and TPs in the SBZ because applying the trigonometry double-angle formula to the Hamiltonian (1) indicates that the minimal SBZ is only one-eighth of the entire SBZ. **c–e** Real-eigenfrequency Riemann surfaces in the $\phi_2 = \pi/2$ plane (**c**), $\phi_3 = \pi/2$ plane (**d**), and $\phi_3 = \pi/3$ plane (**e**). The EP3s and EL3 are denoted by the red stars and dashed curves, respectively. The blue dashed curves show the EL2, which are cuts of the ES2.

$\mathcal{W} = \sum_{\mu \neq \nu} [-\frac{1}{2\pi} \oint_{S^1} (d\vec{\phi} \cdot \nabla_\phi \arg(\omega_\mu - \omega_\nu))]$ with $\mu, \nu = 1,2,3$ indexing the states, which is a topological invariant for the spectral topology of the eigenvalue manifold. The result is $\mathcal{W} = 0$. Apparently, both methods "enclose" the ES2 and EL3 together, therefore they cannot reveal the topological details carried by the ES2 or EL3 individually.

Hence we need to find an alternative approach to characterize the topology of the EL3. We begin by re-examining the fundamentals of the well-established approaches that characterize the topology of EPs. One approach is by counting the winding of the discriminant $\mathcal{D}$ ($:= \text{Re}(\mathcal{D}) + i\text{Im}(\mathcal{D})$), which gives an invariant called the discriminant number (DN). The DN captures the intersection multiplicity of $\text{Re}(\mathcal{D}) = 0$ and $\text{Im}(\mathcal{D}) = 0$ since it reflects the sense of rotation for the discriminant fields $\nabla(\arg \mathcal{D})$[23]. For an EP2 in a two-state system, the winding of the discriminant fields is equivalent to $\nabla_\phi \arg[(\omega_\mu - \omega_\nu)^2]$ – the latter directly leads to eigenvalue winding number $\mathcal{W}$ that captures the topology of the EP2 and EL2[23]. But for systems with more than two states, e.g., the three-state system in this work, the condition $\mathcal{D} = 0$, which is commonly used as the condition for identifying an EP$n$, merely indicates two or more identical eigenvalues. However, this condition does not distinguish EPs of different orders $n$. It follows that what we need is a quantity that vanishes only at the EP3 but is insensitive to any EP2. To this end, we observe that our three-state Hamiltonian has three different resultants, $\mathcal{R}[p, p^{(1)}]$, $\mathcal{R}[p, p^{(2)}]$, and $\mathcal{R}[p^{(1)}, p^{(2)}]$. The fact that $\mathcal{D} = -\mathcal{R}[p, p^{(1)}]$ rules out $\mathcal{R}[p, p^{(1)}]$, and the other two resultants, $\mathcal{R}[p, p^{(2)}]$ and $\mathcal{R}[p^{(1)}, p^{(2)}]$, fulfill our needs. We then define a vector field as $\Lambda(\vec{\phi}) := \eta + i\zeta$, with

$\eta = \mathcal{R}[p^{(1)}, p^{(2)}], \zeta = \mathcal{R}[p, p^{(2)}]$. This way, $\Lambda$ vanishes only at EP3 and completely ignores EP2 (Supplemental Materials). Such a choice is proper and unique and can capture the topology of EL3 by establishing the connection between the intersection multiplicity and the resultant field[50].

In Fig. 4b, we plot the $\nabla_\phi \text{Im}(\ln \Lambda)$ as a vector field (dubbed the resultant field or $\Lambda$-field) on the $\phi_1 \phi_2$-plane at $\phi_3 = 0.7\pi$ (the green plane in Fig. 4a), which intersects with two EL3. $\Lambda$ is indeed vanishing at the EP3, but it does not generate any vortex. Protected by the PT symmetry, the resultants $\eta$ and $\zeta$ purely real, and the topology embedded in the $\Lambda$-field can be described by the winding numbers of $\Lambda$, defined as

$$\mathcal{W}_\Lambda = -\frac{1}{2\pi} \oint_{S^1} \nabla_\phi(\arg \Lambda) \cdot d\vec{\phi}. \quad (2)$$

$\mathcal{W}_\Lambda$ for both EP3 (red stars in Fig. 4a) are zero, suggesting that they both are unstable. To further reveal the local evolution of the EP3, we introduce two types of symmetry-preserving perturbations ($\delta_L$ and $\delta_M$): $L(\phi_1) = -60.68(0.50 \sin^2 \phi_1 - 1 + \delta_L)$ and $M(\phi_2) = -38.62(\cos^2 \phi_2 + \delta_M)$. When the perturbation is off, i.e., $\delta_M = \delta_L = 0$, the two surfaces defined by $\eta = 0$ and $\zeta = 0$ are respectively shown by orange and blue surfaces in Fig. 4a. According to Bezout's theorem, the number of intersection points of two algebraic curves, including points at infinity, is determined by the product of their degrees[50]. In

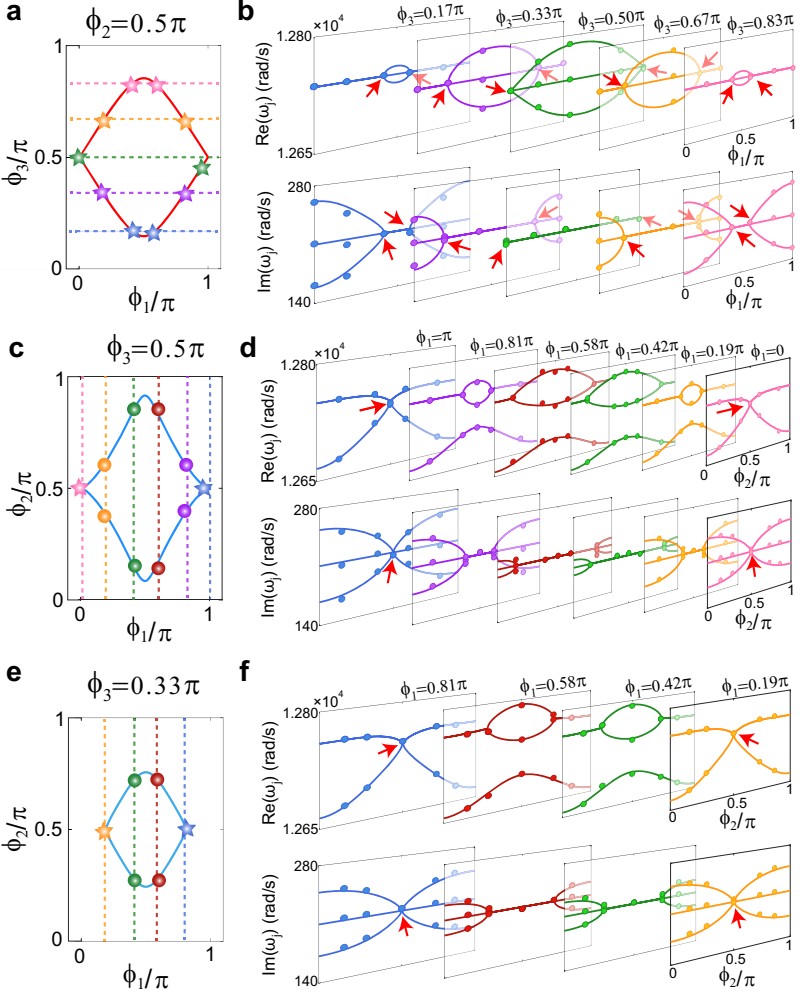

**Fig. 3 | Observation of EL3 and ES2. a** The EL3 in the $\phi_2 = 0.5\pi$ plane. **b** The measured real (upper panels) and imaginary (lower panels) parts of eigenfrequencies along the dashed lines in **a**. An EP3 occurs when both real and imaginary parts of three eigenfrequencies coalesce. **c–f** Two slices of the ES2 at **c** $\phi_3 = 0.5\pi$ and **e** $\phi_3 = 0.33\pi$. The measured real and imaginary parts of eigenfrequencies along the dashed lines in **c** and **e** are shown in **d** and **f**, respectively. The circles in **b**, **d**, and **f** are experimental results. The stars and circles in **a**, **c**, and **e** denote the observed positions of EP3 and EP2. The red arrows in **b**, **d**, and **f** point at the EP3s.

our case, the number of intersection points is four in $\phi_1\phi_2$-plane when considering both the complex domain and intersection multiplicity (Supplemental Materials). However, in Fig. 4b, only two intersections are found, which are the EP3s. This indicates a two-fold multiplicity for both intersections. Further changing $\phi_3$ to $\pi/2$, two EL3 merge and form the TP, which clearly has a multiplicity of four.

The two-fold multiplicity of the EL3 combined with their vanishing $\mathscr{W}_\Lambda$ together suggests that the EL3 in Fig. 4a can be made locally stable without breaking the symmetry (Supplemental Materials). This is verified by letting either $\delta_M$ or $\delta_L$ be non-zero. Figure 4c shows that when $\delta_M = -0.1$, the EL3 split into two pairs symmetric about the $\phi_2 = \pi/2$ plane, and they do so without dropping the order. Figure 4d plots the $\Lambda$-field and the solutions for $\eta = 0, \zeta = 0$ in the $\phi_1\phi_2$-plane at $\phi_3 = 0.7\pi$. Clearly, four EP3s are seen, indicating the removal of multiplicity. The EP3s can be separated into two pairs by the opposite vortices they carry, indicating $\mathscr{W}_\Lambda = \pm 1$, which means they are topologically stable. Note that the TPs from the crossings of the two oriented order-3 ELs possess zero $\mathscr{W}_\Lambda$, and their multiplicity is reduced to two. Based on the winding of $\Lambda$, we can assign each EL3 with a "topological current" using the right-hand rule, as indicated by the arrows in Fig. 4c. Indeed, the currents cancel when the two pairs of EL3 merge at $\delta_M = 0$. When $\delta_M$ is increased to positive, the EL3 vanishes from our

system. In other words, the topological currents defined by the winding of $\Lambda$ are able to capture the merging and annihilation of the EL3. The topological currents are also informative in revealing the local evolution of the TPs. Such a configuration discloses two possible local evolutions in the natural projective plane ($\phi_1\phi_3$ plane). When the TP is open, the two linear-crossed EL3s can only separate without violating the orientation defined by the currents. The two possible cases are shown in Fig. 4e, f.

## Discussion

We compare different choices of the resultants in order to further digest the relationship between the resultant fields and the topology of EP$n$. As mentioned before, if the goal is to only determine the location of EP$n$ in an $n$-level non-Hermitian system, there can be multiple choices of resultants $\mathscr{R}[p^{(j)}, p^{(i)}]$ with $0 \leq j, i < n - 1$ and $j \neq i$. Figure 5a–c plot the three possible choices of resultant fields for our model [Eq. (1)]: $\Lambda = \eta + i\zeta$ in Fig. 5a (the one in focus in our work), $\Lambda' = \chi + i\zeta$ in Fig. 5b, and $\Lambda'' = \chi + i\eta$ in Fig. 5c, with $\chi = \mathscr{R}[p, p^{(1)}] \; (= -\mathscr{D})$. However, as we will show next, only $\Lambda$ is the proper resultant field.

First of all, in all three choices, EP3s can be identified as the intersection of the curves at which the real and imaginary parts of the resultant fields vanish. However, the winding of the resultant fields

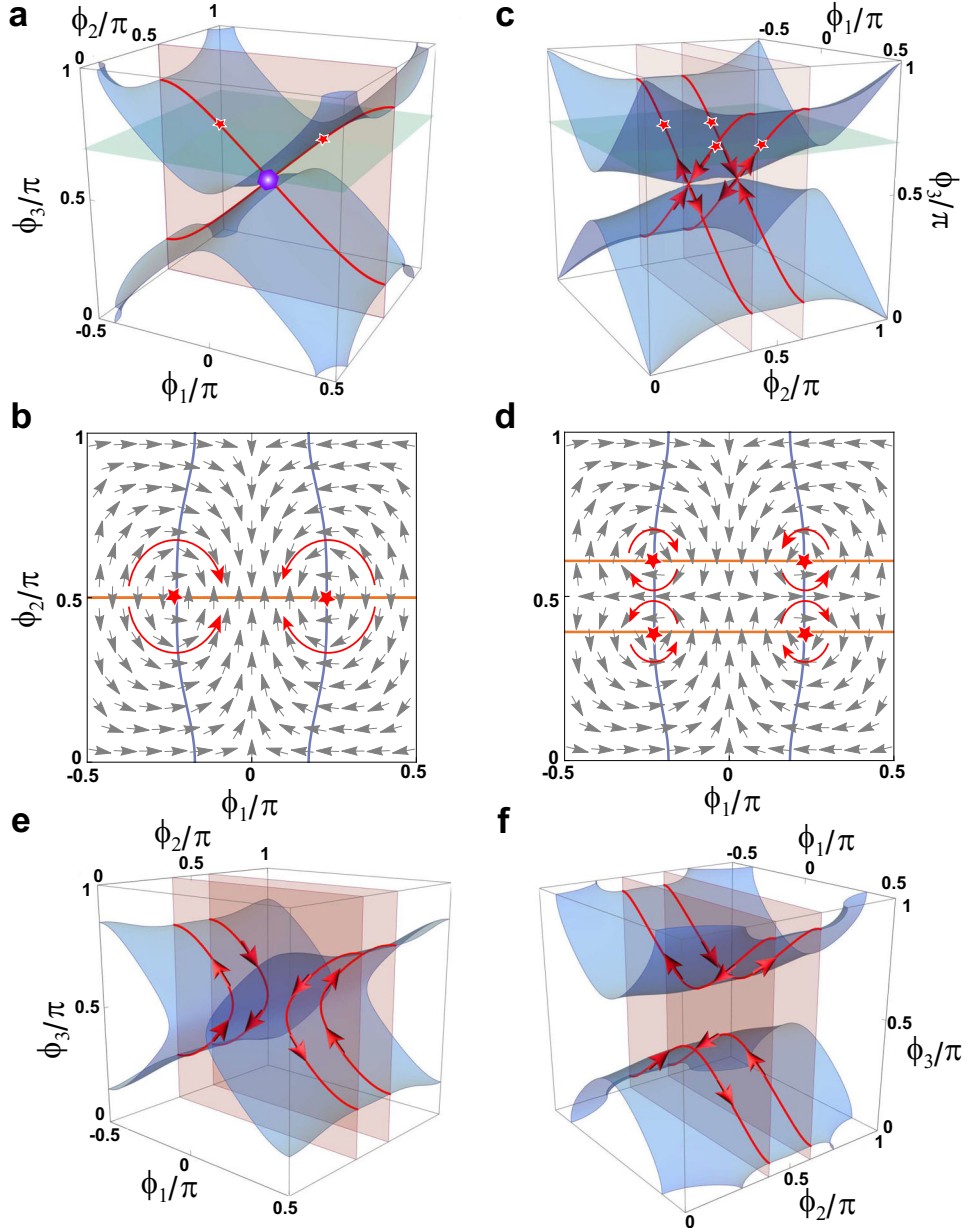

**Fig. 4 | Topological characterization of EP3 and EL3. a** Topologically neutral EL3. The blue and red surfaces stand for Re($\Lambda$) = 0 and Im($\Lambda$) = 0. The purple dot is the TP. **b** The $\Lambda$-field (arrows) in $\phi_3 = 0.7\pi$ plane [the green plane in **a**]. The EP3 (red stars) are found at the intersections of the blue line [Re($\Lambda$) = 0] and the orange line

[Im($\Lambda$) = 0]. **c** With $\delta_L = 0$ and $\delta_M = -0.1$. The EL3 split into two pairs with opposite topological currents. **d** The $\Lambda$-field in the $\phi_3 = 0.7\pi$ plane. **e, f** The opening of the TPs under different perturbations: **e** $\delta_L = -0.1$, $\delta_M = -0.1$; **f** $\delta_L = 0.1$, $\delta_M = -0.1$. The red arrows in **b, d** are guides to the eye.

around the EP3s may not match the intersection multiplicity of the corresponding resultants. For example, in Fig. 5b, c, cusps are seen on the green curves indicate $\chi = \mathscr{R}[p, p^{(1)}] = 0$. A cusp carries an intersection multiplicity of 2 because it can be viewed as a self-intersection, and the net multiplicity at the EP3s is 3 and 2 when diagnosed using $\Lambda'$ and $\Lambda''$. However, the winding of $\Lambda'$ and $\Lambda''$ around the EP3 is 1 and 0, respectively (Fig. 5e, f), which clearly does not match the multiplicity. The reason for this failure is because the condition $\chi = \mathscr{R}[p, p^{(1)}] = 0$ is satisfied as long as two roots of the discriminant are equal, hence $\chi = 0$ contains singularities in itself (EP2s), and thus the connection between intersection multiplicity and resultant winding number fails. In comparison, the resultant field by $\Lambda$ detects only EP3, and its winding around the EP3 matches the intersection multiplicity, as shown in Fig. 5a, d. The detailed calculation of the intersection multiplicity is presented in the Supplementary Information.

The resultant field approach can be generalized to characterize EP$n$ with $n > 3$. The choice of resultants simply needs to exclude all the resultants already used in the lower-order EPs. For example, Fig. 5g depicts the resultant choice for EP4. Explicitly, the proper choice for the EP$n$ is $\mathscr{R}[p^{(j)}, p^{(n-1)}]$ with $0 \le j < n - 1$. The topological description of the EP$n$ then becomes the problem of characterizing the $(n-1)$-component complex vector. And in the presence of additional symmetry, such as the PT symmetry here, the problem further reduces to characterizing the $(n-1)$-component real vector.

The EL3 demonstrated here together with previous works show that the higher-order EPs possess far richer topological properties that are absent for both EP2 and Hermitian degeneracies. The hybrid topological winding number and the associated fractional Berry phases have been demonstrated to be a unique feature of higher-order EPs using the eigenvectors[36,46]. Within the context of non-Hermitian bands,

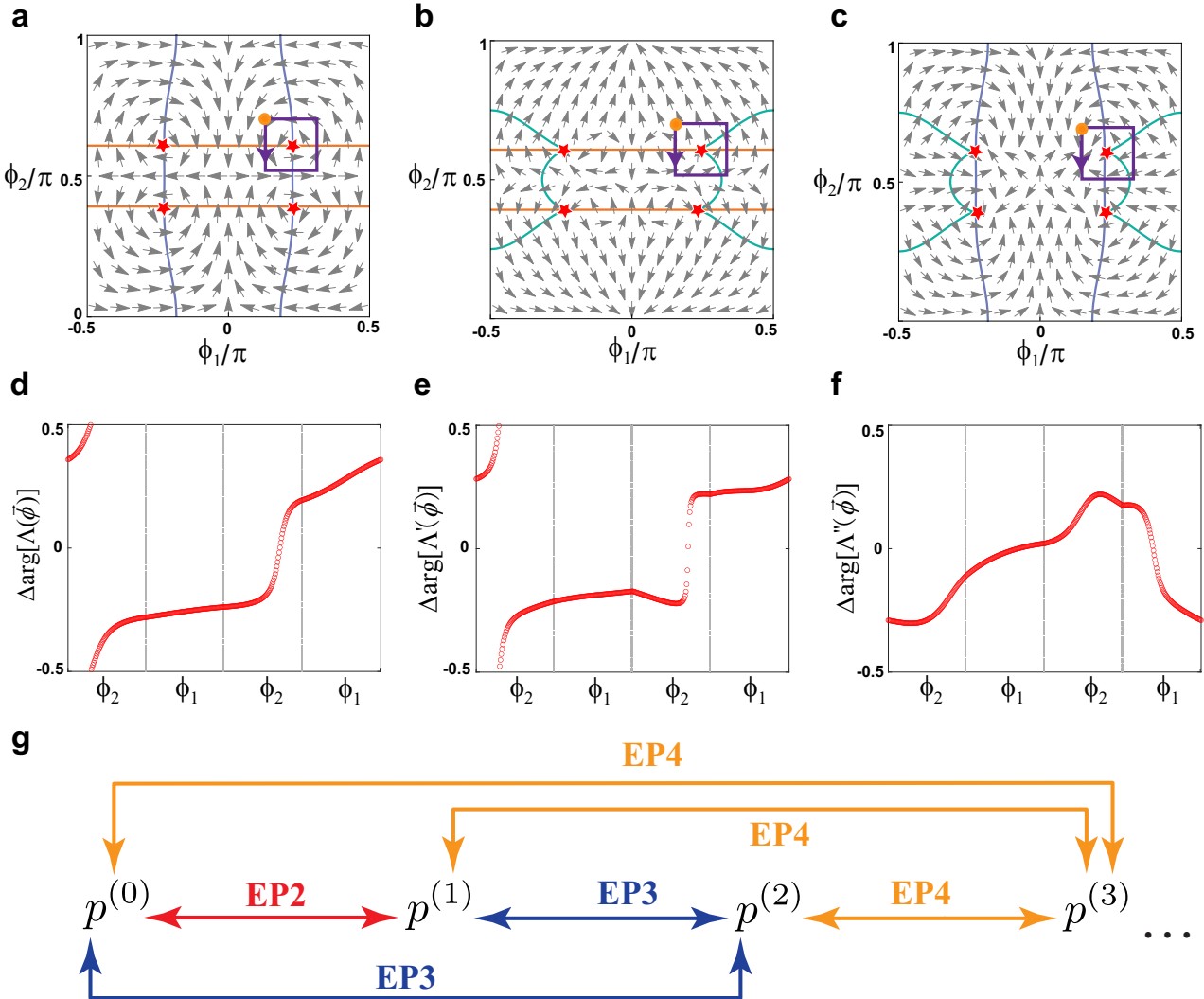

**Fig. 5 | Intersection multiplicity and the proper resultant field for the EP3s.**
**a**–**c** show the three possible choices of resultant fields (arrows). The plots are in the $\phi_1\phi_2$-plane at $\phi_3 = 0.7\pi$, denoted by the green plane in Fig. 4c. Here, the quantity to define the resultant fields is $\Lambda = \eta + i\zeta$ in **a**, $\Lambda' = \chi + i\zeta$ in **b**, and $\Lambda'' = \chi + i\eta$ in **c**. **d**–**f** The corresponding evolutions of the argument when encircling the EP3s along the purple loop. **a**–**c** The orange, blue, and green curves respectively represent $\zeta = 0$, $\eta = 0$, and $\chi = 0$. The red stars denote EP3s. The orange points indicate the start and finish points, with the purple arrows denoting the direction of encircling. Note that **a** is identical to Fig. 4d. It is reproduced here for the convenience of comparison. **g** Schematics for choosing resultants to characterize the EP$n$. The red, blue, and orange arrows denote the resultants chosen for the EP2, EP3, and EP4, respectively. $p^{(j)}$ is the $j$th-order derivative of the characteristic polynomial with respect to $\omega$.

the higher-order EPs serve as the cusp singularities of multiple EL2s in the 3D space[46], and the topological characterization of EL2s viewing from the eigenvalue manifold necessitates the braid group[51,52], giving rise to the eigenvalue knots[20,21,53] and non-Abelian conservation rule[47,54]. Such the fact that the EL2s possess much more fruitful topological properties than the single EP2s also holds for the higher-order ELs, but the approach applied to the EL2s fails in the higher-order ELs. Our work here uncovers that EPs of different orders may form structures that challenge the conventional wisdom of topological characterization, and they necessitate an auxiliary resultant manifold, which remains well-behaved at the EP2 and only detects the EP3. Although the EL3 in this work is embedded in the ES2 originate from a single band gap, the resultant manifold approach can not only apply to the EL3 intersected by the ES2 from adjacent band gaps[55] but also be generalized to higher-order ELs (Supplemental Materials), which paves the way to digest the topology of higher-order ELs in higher-dimensional non-Hermitian bands. The exploration of these properties may lead to new phenomena and applications relating to non-Hermitian energy transfer[56,57] or wave manipulations[58].

## Data availability
The data that generate the results of this study are available from the corresponding authors upon request.

## Code availability
The codes supporting the findings of this study are available from the corresponding authors upon request.

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

## Acknowledgements

This work is supported by the National Key R&D Program of China (No. 2022YFA1404500, 2022YFA1404400, 2022YFA1404701), the National Natural Science Foundation of China (12174072, 2021hwyq05), the Hong Kong Research Grants Council (RFS2223-2S01, 12302420, 12301822), and the Natural Science Foundation of Shanghai (No. 21ZR1403700).

## Author contributions

G.M. and K.D. conceived the idea. G.M. and W.T. designed the experiments. W.T. conducted numerical simulations, analytical calculations, and experiments. G.M., K.D., and W.T. participated in the discussions and analyses of the results. G.M. and K.D. supervised the project. All authors contributed to the composition of the paper.

## Competing interests

The authors declare no competing interests.
