## [Peer Review File · Nature Communications]

Realization and Topological Properties of Third-Order Exceptional Lines Embedded in Exceptional SurfacesREVIEWER COMMENTS

Reviewer #1 (Remarks to the Author):

The Authors report an experimental realization of third-order exceptional lines embedded in second-order exceptional surfaces in three-dimensional synthetic space of an acoustic cavity. Specifically, the Authors faithfully measure the complex spectrum and its unique topological structure in synthetic space. The Authors further propose a theoretical characterization of the third-order exceptional lines based on the resultant winding number.

Recently, the physics of non-Hermitian Hamiltonians has attracted growing interest. In this context, exceptional points give rise to a number of physical phenomena unique to non-Hermitian systems and play an important role both in theory and experiments. Thus, I believe that this manuscript, which experimentally realizes a new type of nodal structure of exceptional points unique to non-Hermitian systems, can make a significant contribution in non-Hermitian physics.

However, the validity of several important descriptions in this manuscript is not well elaborated on. Additionally, the novelty compared with the literature is unclear. Before I make a decision, I would like to request that the Authors address the following concerns.

(1) I find the precise definitions of "EL3" and "ES2" unclear and ambiguous, which should be clarified at the beginning of the manuscript. For example, it is unclear to me whether the Authors define "EL3" as merely the spectral degeneracy of the three eigenstates or additionally require the rank deficit of 2 (i.e., the rank of the 3-by-3 Hamiltonian is 1 there). In fact, the mere degeneracy of the three eigenstates does not necessarily determine the rank deficit.

Similarly, "ES2" in this manuscript seems to be the same as the symmetry-protected exceptional surface in three dimensions well discussed in the literature [e.g., R. Okugawa & T. Yokoyama, Phys. Rev. B 99, 041202(R) (2019); H. Zhou et al., Optica 6, 190 (2019); K. Kawabata et al., Phys. Rev. Lett. 123, 066405 (2019)]. If so, the Authors should clarify this in the manuscript.

(2) According to the abstract (especially, on Lines 19-23), the theoretical characterization of the third-order exceptional line based on the resultant winding number seems like one of the important results found in the present manuscript for the first time. However, on Line 54, the manuscript reads, "This conundrum is resolved by exploring the resultants of the Hamiltonian matrix [38], ...". In fact, I find a similar analysis in Ref. [38] [i.e., P. Deplace et al., Phys. Rev. Lett. 127, 186602 (2021)], as well as several other related works such as Ref. [39] [i.e., I. Mandal & E. J. Bergholtz, Phys. Rev. Lett. 127, 186601 (2021)]. Thus, I would like to request that the Authors clarify the novelty of their theoretical analysis compared with these previous works. If it is not original, the Authors should put more emphasis on the previous works.

(3) On Line 66, I fail to clearly understand the meaning of "An isolated EP_n requires 2(n-1) degrees of freedom (DOFs) to have a solution", as well as the reason for this statement. The Authors should clarify it.

(4) On Line 68, while the Authors cite Refs. [33, 41, 43] for the references on exceptional points in a four-dimensional parameter space, I cannot find any discussions on them in these references.

(5) In Eq. (1), the Authors introduce their model. I would like to request that the Authors clarify the actual physical origin of non-Hermiticity in their model. According to the manuscript, it seems to be a kind of dissipation, but more detailed explanations are needed. Although the Authors provide an explanation on Lines 98-100, I believe that this is a possible interpretation in synthetic space, and the actual physical meaning is unclear. While the Authors may provide a detailed explanation in the supplementary information, they should explain it also in the main text.

(6) On Line 109, the Authors introduce "touching points (TPs)". I would like to request that the Authors clarify whether they define the "touching points (TPs)" merely by the crossing points

between "EL3" (and hence "touching points" are literally touching points and do not contain any special meanings) or they require an additional nontrivial property as a defining property.

(7) On Line 120, the Authors explain their experimental results, "The coalescence of two of the three states or all these states is clearly seen (Fig. 3d, f)". While I can indeed see the coalescence of two or three states in Fig. 3 (d, f), I fail to clearly understand whether or not this coalescence corresponds to the third-order exceptional point solely from these figures (i.e., the order of exceptional points is unclear solely from these figures). The Authors should add more explanations.

(8) In the section "Characterization of the EL3", while the Authors discuss the topological characterization of "EL3", they do not discuss that of "ES2". Since the stability of "EL3" and "ES2" seems to be guaranteed independently, the Authors should also discuss the topological characterization of "ES2". It seems to me that it is guaranteed by the Z_2 topological invariant protected by parity-time symmetry, introduced in several previous works [e.g., R. Okugawa & T. Yokoyama, Phys. Rev. B 99, 041202(R) (2019); H. Zhou et al., Optica 6, 190 (2019); K. Kawabata et al., Phys. Rev. Lett. 123, 066405 (2019)], which the Authors should clarify.

(9) In the first paragraph of the section "Characterization of the EL3", the Authors argue that the winding number of the complex spectrum is irrelevant to the stability of "ES2" and "EL3". However, while the Authors claim that "the result is $W=0$ ", it is unclear how they choose the complex eigenvalues (i.e., μ and ν) and the one-sphere S^1 . The Authors should clarify whether they obtain $W=0$ for all the possible choices of μ , ν , S^1 or only for some specific choices.

(10) In the section "Characterization of the EL3", the Authors introduce the resultant winding number in Eq. (2) that guarantees the stability of "EL3". There, the Authors sometimes emphasize the role of symmetry. For example, the manuscript reads, "we introduce two types of symmetry-preserving perturbations" (Line 151) and "... the EL3 in Fig. 4a can be made locally stable without breaking the symmetry". Here, I fail to clearly understand what symmetry they focus on and hence would like to request that the Authors specify which symmetry they consider.

Furthermore, if the Authors are to believe that symmetry protection is relevant to the stability of their "EL3", such symmetry should be necessary for the introduction of the topological invariant in Eq. (2). However, the role of symmetry in Eq. (2) is unclear in the present manuscript; the Authors should clarify it.

In other words, if symmetry protection is indeed relevant to the stability of the Authors' "EL3", "EL3" should disappear immediately if symmetry-breaking perturbations are added. The Authors should explicitly confirm this by adding such symmetry-breaking perturbations.

(11) On Line 154, while the Authors use "Bezout's theorem", they do not provide any explanations about it. The Authors should clarify what the statement of "Bezout's theorem" is and why it is relevant here.

(12) On Line 177, the manuscript reads, "The EL3 demonstrated here together with previous works [34, 44, 48] show ...". However, the relationship between this manuscript and these previous works [34, 44, 48] [i.e., K. Ding et al., Phys. Rev. X 6, 021007 (2016); W. Tang et al., Science 370, 1077 (2020); J. Hu et al. (2022)] is unclear solely from this sentence. Thus, I would like to request that the Authors clarify the similarities and differences with these previous works in more detail.

Reviewer #2 (Remarks to the Author):

Realization and Topological Properties of Third-Order Exceptional Lines Embedded in Exceptional Surfaces

By Weiyuan Tang, Kun Ding, Guancong Ma

In this work, authors report the realization of order-3 exceptional lines in coupled acoustic cavities.

However, I could not find the significance of this paper because a similar phenomenon is already reported in previous works.

Weiyuan Tang Science 370 6520 (2020).

Kun Ding, Phys. Rev. X 6, 021007 (2016).

Also, I could not find the novelty of the theoretical part because it is essentially the same as the discussion in Ref. [43] and Ref. [34].

For the above reason, I recommend the publication in another journal such as Scientific Reports.

Reviewer #3 (Remarks to the Author):

In the manuscript "Realization and Topological Properties of Third-Order Exceptional Lines Embedded in Exceptional Surfaces", the authors report both experimental and theoretical findings. Experimentally, they realize EP3 Line embedded in an EP2 surface within a three-dimensional periodic parameter space, utilizing coupled acoustic cavities. Theoretically, they propose the winding number of resultants to pinpoint the topological current of EP3 line embedded in an EP2 surface, since the winding number of eigenvalues is ill-defined in this situation.

The most remarkable aspect of their findings is the proposed resultant winding number, which enables the analysis of the topological charge and evolution of higher-order ELs when they are embedded in a lower-order EP manifold. I think this is an inspiring work, potentially offering significant contributions to the relevant research fields, especially PT-symmetric non-Hermitian systems. I can thus recommend it for publication in Nature Communications if the following points are clarified.

(1) The authors should clearly show the expression of discriminant D at its first appearance in the manuscript. Readers may not be familiar with this formula.

(2) The authors present the EP2 surface, EP3 line, and touching point in Fig.2(b) in an 8-fold larger synthetic Brillouin zone. The figure looks unclear and may hide the information that the touching point is only one point at the boundary of SBZ. I think all features could be clearly shown within a single SBZ. Alternatively, the authors should point out this information in the figure caption.

(3) In Fig. 3(a) and 3(c), the evolution of imaginary energy becomes crucial when the touching point is traversed in different parameter directions. The authors should show the corresponding evolution of imaginary energy when crossing the touching point in different directions.

(4) Can the method of the resultant field be extended to higher-order EP n where $n > 3$? Is there a general principle for constructing the real and imaginary parts of the resultant vector field?

(5) The authors describe the third-order exceptional line in Fig. 4(a) as "topologically neutral". Is there an additional symmetry contributing to the winding number of resultant field being zero? Could the authors provide further clarification on this "topologically neutral"?

(6) There are a few typographical errors: In the caption of Fig. 4: “the field (arrows) in of”; On page 8 of Supplemental Information: “we define two different resultant fields ...”.

Reply to Reviewer #1

The Authors report an experimental realization of third-order exceptional lines embedded in second-order exceptional surfaces in three-dimensional synthetic space of an acoustic cavity. Specifically, the Authors faithfully measure the complex spectrum and its unique topological structure in synthetic space. The Authors further propose a theoretical characterization of the third-order exceptional lines based on the resultant winding number.

Recently, the physics of non-Hermitian Hamiltonians has attracted growing interest. In this context, exceptional points give rise to a number of physical phenomena unique to non-Hermitian systems and play an important role both in theory and experiments. Thus, I believe that this manuscript, which experimentally realizes a new type of nodal structure of exceptional points unique to non-Hermitian systems, can make a significant contribution in non-Hermitian physics.

Response: We sincerely thank the referee for evaluating our realization of the third-order exceptional lines embedded in the second-order exceptional surfaces as a significant contribution to non-Hermitian physics. The following are the responses to the suggestions and comments.

However, the validity of several important descriptions in this manuscript is not well elaborated on. Additionally, the novelty compared with the literature is unclear. Before I make a decision, I would like to request that the Authors address the following concerns.

(1) I find the precise definitions of “EL3” and “ES2” unclear and ambiguous, which should be clarified at the beginning of the manuscript. For example, it is unclear to me whether the Authors define “EL3” as merely the spectral degeneracy of the three eigenstates or additionally require the rank deficit of 2 (i.e., the rank of the 3-by-3 Hamiltonian is 1 there). In fact, the mere degeneracy of the three eigenstates does not necessarily determine the rank deficit.

Similarly, “ES2” in this manuscript seems to be the same as the symmetry-protected exceptional surface in three dimensions well discussed in the literature [e.g., R. Okugawa & T. Yokoyama, Phys. Rev. B 99, 041202(R) (2019); H. Zhou et al., Optica 6, 190 (2019); K. Kawabata et al., Phys. Rev. Lett. 123, 066405 (2019)]. If so, the Authors should clarify this in the manuscript.

Response: We thank the referee for raising this point. The “EL3” in our work refers to the rank-deficit-of-2 case instead of a mere spectral degeneracy. There are two types of EPs in non-Hermitian systems, the non-defective and defective ones. The non-defective EPs, a straightforward extension of the Hermitian degeneracy point, are degeneracies in eigenvalues, while the associated eigenvectors are still orthogonal. In contrast, a defective EP only emerges when both the eigenvalues and eigenvectors become degenerate, which has no counterpart in Hermitian systems.

In our manuscript, the EP always refers to the defective one, and thereby the EL3 (ES2) is constituted of order-3 (order-2) defective EPs.

The ES2 in our work is PT symmetry-protected exceptional surface in three dimensions. The referee mentioned three papers: *R. Okugawa & T. Yokoyama, Phys. Rev. B 99, 041202(R) (2019)*; *H. Zhou et al., Optica 6, 190 (2019)*; *K. Kawabata et al., Phys. Rev. Lett. 123, 066405 (2019)*. The PRB and PRL papers together give the topological classification of the order-2 EP geometry, e.g., points, lines, and surfaces, in d -dimensional parameter spaces but have not considered and involved higher-order EPs. According to Table 1 in the PRL paper, our ES2, with codimension 1 under the point gap, is a \mathbb{Z}_2 classification. However, the main advancement of our work is the characterization of the EL3. Our study clearly shows that higher-order EPs can give rise to new, exotic topology that does not exist in two-level systems, therefore, goes beyond the scope of the PRB and PRL papers.

The Optica paper presents a numerical model of ES2 produced from nodal-line semimetal, but there is no discussion on its topology.

Also, we would like to emphasize that our work also contains experimental components, whereas the three papers mentioned here are purely theoretical.

In the revised main text, besides citing Phys. Rev. Lett. 123 as Ref. [20], Optica 6, 190 as Ref. [32], and Phys. Rev. B 99 as Ref. [33], at both the end of the first paragraph and the beginning of the second paragraph, we have clarified the definitions of EL3 and ES2 more explicitly as

“Non-Hermitian degeneracies known as EPs possess topological properties characterizable by spectral winding numbers^{6,8,9,20–22}. Most studies focus on EPs formed by two coalescing states with one being defective, i.e., defective order-2 EPs. Akin to Hermitian degeneracies, they can also form nodal structures, such as rings^{23–27}, lines^{28–30}, links and chains^{28,30,31}, and surfaces^{20,32,33}. Higher-order EP is formed when three or more states coalesce, with two or more states being defective.”

“The EL3 (ES2) is formed by order-3 (order-2) defective EPs.”

(2) According to the abstract (especially, on Lines 19-23), the theoretical characterization of the third-order exceptional line based on the resultant winding number seems like one of the important results found in the present manuscript for the first time. However, on Line 54, the manuscript reads, “This conundrum is resolved by exploring the resultants of the Hamiltonian matrix [38], ...”. In fact, I find a similar analysis in Ref. [38] [i.e., P. Deplace et al., Phys. Rev. Lett. 127, 186602 (2021)], as well as several other related works such as Ref. [39] [i.e., I. Mandal & E. J. Bergholtz, Phys. Rev. Lett. 127, 186601 (2021)]. Thus, I would like to request that the Authors clarify the novelty of their theoretical analysis compared with these previous works. If it is not original, the Authors should put more emphasis on the previous works.

Response: We appreciate the referee’s suggestion. Reference [38] is an important work that first proposed using the resultant vector to characterize the EP3 but cannot be applied to predict the possible local evolutions of EL3s investigated in our work. Hence, we do wish to emphasize several essential developments compared to Ref. [38].

- First of all, the critical difference between our work and Ref. [38] is the selection of resultants associated with the calculation of resultant winding number, which is crucial for revealing the topology of EP n . In an n -level non-Hermitian system, there can be multiple choices of resultants $\mathcal{R}[p^{(j)}, p^{(i)}]$ with $0 \leq j, i < n - 1$ and $j \neq i$ to determine the EP n . The choice of resultants in Ref. [38] is $\mathcal{R}[p^{(j)}, p^{(j+1)}]$ with $0 \leq j < n - 1$. However, the resultants in Ref. [38], $\mathcal{R}[p^{(j)}, p^{(j+1)}]$, can identify the locations of EPs in the parameter space, but it may not be suitable to evaluate their topological properties. To this end, the advancement in our work is in the demonstration that there exists a unique choice of the resultants for revealing the topology of EP n . This choice is made by establishing a connection between the multiplicity of the EP and the resultant winding number. In short, as depicted in Fig. R1, the choice of resultants for EP n must eliminate all the resultants already used in the lower-order EPs, say, the order less than n . Especially in our case, we must avoid using $\mathcal{R}[p, p^{(1)}]$, which is nothing but the discriminant in our three-level system, leading us to define the resultant winding number \mathcal{W}_Λ with $\Lambda = \eta + i\zeta$ and $\eta = \mathcal{R}[p^{(1)}, p^{(2)}], \zeta = \mathcal{R}[p, p^{(2)}]$. By studying the relation between the winding of Λ and the intersection multiplicity of the EP3, we show that this choice of Λ is unique in that it ignores the influence of EP2 and only detects EP3, and is successful in topologically characterizing the EP3s despite the presence of the EP2s. In comparison, Ref. [38] characterizes the EP3 by defining the resultant winding number in terms of $\mathcal{R}[p, p^{(1)}]$ and $\mathcal{R}[p, p^{(2)}]$, which is proved to be inappropriate for the topological characterization of EP3 by our work. This is because $\mathcal{R}[p, p^{(1)}]$ contains singularities in itself (EP2s), so the relation between intersection multiplicity and resultant winding number for the EP3 is not satisfied because the winding must go through EP2s in our case.

From the referee’s comments, we feel that the reasoning behind the choice of the resultant field and its uniqueness is not clearly presented in the original manuscript. We have substantially revised the manuscript (the “Characterization of the EL3” section) to strengthen the relevant discussions and highlight the role of proper resultants. In particular, three paragraphs (in the “Discussion and Conclusions” section) and a new figure (Fig. 5) have been added to the revised main text. We believe these revisions also differentiate our work from Ref. [38]. (Ref. [39] did not mention the topological characterization of EP3s)

FIG. R1. Schematic for the choice of resultants to characterize the EP2 and EP3.

- Secondly, our work here is the first experimental demonstration of not only an EL3 but also non-Hermitian singularities that have to be characterized using resultant winding numbers. In Ref. [38], the EP3 and EL3 are theoretically investigated based on the linearized rotating shallow water model. It is a model that is very interesting in its own right, and it is no doubt an impressive extension of non-Hermitian models to potentially an entirely new realm of physics. However, acoustic platforms are more versatile by offering immense and more detailed controls of system parameters (thanks to the years of development in phononic crystals, metamaterials, and non-Hermitian acoustics), which are essential for realizing sophisticated models in this work.

(3) On Line 66, I fail to clearly understand the meaning of “An isolated EP n requires $2(n-1)$ degrees of freedom (DOFs) to have a solution”, as well as the reason for this statement. The Authors should clarify it.

Response: We thank you for the nice suggestion. The DOFs of an isolated EP n are associated with the number of required equations for a solution. For instance, an EP2 emerges when one equation of complex eigenvalues ($\omega_1 = \omega_2$) is satisfied; an EP3 exists when two equations ($\omega_1 = \omega_2$ and $\omega_2 = \omega_3$) are satisfied; an EP4 requires three equations, namely, $\omega_1 = \omega_2$, $\omega_2 = \omega_3$, and $\omega_3 = \omega_4$. Generally, an EP n occurs when $n-1$ equations of complex eigenvalues are satisfied. Note that the equality of two complex numbers means that both real and imaginary parts are equal. Therefore, an isolated EP n requires $2(n-1)$ degrees of freedom to have a solution. This can also be interpreted from the aspect of characteristic polynomials. For an $n \times n$ effective Hamiltonian H , the characteristic polynomial is $p(\omega) = \det(\omega \mathbf{I} - H) = a_n \omega^n + a_{n-1} \omega^{n-1} + \dots + a_1 \omega + a_0$. At an EP n , $p(\omega)$ and its $n-1$ successive derivatives $p(\omega)^{(j)} \equiv \partial^j p(\omega) / \partial \omega^j$ must vanish. Since $p(\omega)$ and its $n-1$ successive derivatives are complex, both their real and imaginary parts must be nil at the isolated EP n , thereby leading to $2(n-1)$ degrees of freedom. We have clarified the degrees of freedom of an EP n in the revised main text (in the first paragraph of the “Realization of symmetry-protected EL3” section) as

“An isolated EP n is found when n complex eigenvalues become identical, i.e., an EP2 emerges at $\omega_1 = \omega_2$, and an EP3 occurs with $\omega_1 = \omega_2$ and $\omega_2 = \omega_3$. In other words, an isolated EP n is a common solution of a set of $n - 1$ equations to be satisfied^{40,41}, and the existence of such a solution requires $2(n - 1)$ degrees of freedom (DOFs).”

(4) On Line 68, while the Authors cite Refs. [33, 41, 43] for the references on exceptional points in a four-dimensional parameter space, I cannot find any discussions on them in these references.

Response: We apologize for mistaking the references on Line 68. We have cited Phys. Rev. Lett. 123, 237202 (2019) and Science 370, 1077–1080 (2020) as Refs. [43, 44], where Ref. [43] studied an ES2 in a four-dimensional parameter space spanned by x, y, θ, H and Ref. [44] investigated an EP3 as a nexus tuned by four parameters $\delta_f, \delta_A, \delta_g, \delta_B$. The relevant sentence in the main text has been revised to

“Hence both isolated EP3 and ES2 are stable in a four-dimensional (4D) parameter space^{43,44}.”

(5) In Eq. (1), the Authors introduce their model. I would like to request that the Authors clarify the actual physical origin of non-Hermiticity in their model. According to the manuscript, it seems to be a kind of dissipation, but more detailed explanations are needed. Although the Authors provide an explanation on Lines 98-100, I believe that this is a possible interpretation in synthetic space, and the actual physical meaning is unclear. While the Authors may provide a detailed explanation in the supplementary information, they should explain it also in the main text.

Response: We thank the referee for the suggestion. In our experiment, the non-Hermiticity is indeed introduced by adjusting the local dissipative rates at each cavity. This is achieved by inserting specific amounts of acoustic sponge into the coupled acoustic cavities. For a given amount of sponge, the dissipation it induces is dependent on the local acoustic kinetic energy. Then the resonant sound field’s spatial profile is leveraged to make the dissipative rate a function of the azimuthal position ϕ_1 .

We agree with the referee that more description of the experimental implementation shall be included in the main text. The discussion initially in the Supplementary Information has been merged with those in the revised main text. We have also moved Table S1, which summarizes the implementation of all three synthetic coordinates, to the main text. Figure S2 is also merged with Fig. 1. For convenience, we have also reproduced it here as Table R1.

TABLE R1. Experimental implementations of the synthetic coordinates.

Synthetic coordinate	Affected quantity	Physical mechanism	Function relation	Implementation
ϕ_1	Onsite dissipative rate (non-Hermiticity)	Dissipation proportional to local kinetic energy, $v^2(\phi)$.	$v^2(\phi_1) \propto \cos^2(\phi_1)$	Variation of the azimuthal position ϕ_1 of the sponge.
ϕ_2	Onsite resonant frequency	Resonant frequency sensitive to the local pressure intensity of $P^2(\phi)$.	$P^2(\phi_2) \propto \sin^2(\phi_2)$	Variation of the azimuthal position ϕ_2 of the metal block.
ϕ_3	Hopping	Coupling strength sensitive to the local pressure intensity $P^2(\phi)$.	$P^2(\phi_3) \propto \sin^2(\phi_3)$	Variation of the azimuthal position ϕ_3 of the coupling holes.

(6) On Line 109, the Authors introduce “touching points (TPs)”. I would like to request that the Authors clarify whether they define the “touching points (TPs)” merely by the crossing points between “EL3” (and hence “touching points” are literally touching points and do not contain any special meanings) or they require an additional nontrivial property as a defining property.

Response: We thank the referee for pointing out this vagueness. Indeed, “touching points (TPs)” only means the crossing points of two EL3s and is a generalization of its definition of nodal lines.

The same term has also been seen in previous literatures, such as Z. Yang, *et al.*, Phys. Rev. Lett. 124, 186402 (2020) as “a touching point (TP), where two nodal lines touch together.” To avoid confusion, we have clarified its meaning when the “touching points (TPs)” first appear in the revised main text as

“The TPs are previously defined to the point where two nodal lines touch⁴⁸, and here we generalize it to EL3.”

(7) On Line 120, the Authors explain their experimental results, “The coalescence of two of the three states or all these states is clearly seen (Fig. 3d, f)”. While I can indeed see the coalescence of two or three states in Fig. 3 (d, f), I fail to clearly understand whether or not this coalescence corresponds to the third-order exceptional point solely from these figures (i.e., the order of exceptional points is unclear solely from these figures). The Authors should add more explanations.

Response: We are grateful for the referee’s suggestion. An EP3 emerges when both real and imaginary parts of three eigenfrequencies coalesce, so it is true that real parts themselves cannot identify the EP3. To explicitly demonstrate the EP3, we have added the measured imaginary parts of eigenfrequencies in the revised Fig. 3b, d, and f. The positions of EP3s are marked by the red arrows therein. To further help the referee to identify the EP3s, we select some panels and plot them in Fig. R3 without the projection perspective.

FIG. R3 Observation of EP3. **a** The measured real and imaginary parts of eigenfrequencies as functions of ϕ_1 with $\phi_2 = 0.5\pi$ and $\phi_3 = 0.33\pi$. **b-c** The measured real and imaginary parts of eigenfrequencies as functions of ϕ_2 with $\phi_3 = 0.5\pi$, $\phi_1 = \pi$ and $\phi_3 = 0.33\pi$, $\phi_1 = 0.19\pi$, respectively. The EP3s occur when both the real and imaginary parts of three eigenfrequencies coalesce. The circles are experimental results. The red arrows point at the EP3s.

(8) In the section “Characterization of the EL3”, while the Authors discuss the topological characterization of “EL3”, they do not discuss that of “ES2”. Since the stability of “EL3” and “ES2” seems to be guaranteed independently, the Authors should also discuss the topological characterization of “ES2”. It seems to me that it is guaranteed by the Z_2 topological invariant protected by parity-time symmetry, introduced in several previous works [e.g., R. Okugawa & T. Yokoyama, *Phys. Rev. B* 99, 041202(R) (2019); H. Zhou et al., *Optica* 6, 190 (2019); K. Kawabata et al., *Phys. Rev. Lett.* 123, 066405 (2019)], which the Authors should clarify.

Response: We appreciate the referee’s nice suggestion. Similar to EL3, the required degrees of freedom of the stable ES2 are diminished from 4 $[2(n - 1) + 2]$ to 3 $[(n - 1) + 2]$ when the system respects PT-symmetry. Thus the ES2 is locally stable in the 3D parameter space, which is also guaranteed by the Z_2 topological invariant [33,20], defined as $s = \text{sgn det}(H_{3b}) = (-1)^{n_{R-}}$ with n_{R-} denoting the number of the real negative eigenvalues when the eigenvalues of H_{3b} are nonzero. We calculate the Z_2 topological invariant of the ES2 in Fig. 2b and obtain the result $s = 1$. Following the referee’s suggestion, besides citing *Phys. Rev. B* 99, 041202(R) (2019) as Ref. [33] and *Phys. Rev. Lett.* 123, 066405 (2019) as Ref. [20], we have discussed the topological characterization of ES2 in the revised main text (the first paragraph in the “Characterization of the EL3” section) as

“The topological properties of ES2 protected by PT-symmetry are characterized by a Z_2 topological invariant^{20,33}, which is equal to 1 here.”

“Yet it remains possible to encircle both the ES2 and EL3 together using the zeroth or first homotopy group⁴⁷. Under the zeroth homotopy group count equivalence classes of 0-sphere, i.e., two separate points, the ES2 and EL3 together form a manifold that is Z_2 classified^{20,33}.”

However, the work, H. Zhou et al., *Optica* 6, 190 (2019) (in the reference list as Ref. [32]), does not discuss the topological characterization of ES2.

(9) In the first paragraph of the section “Characterization of the EL3”, the Authors argue that the winding number of the complex spectrum is irrelevant to the stability of “ES2” and “EL3”. However, while the Authors claim that “the result is $W=0$ ”, it is unclear how they choose the complex eigenvalues (i.e., ω_μ and ω_ν) and the one-sphere S^1 . The Authors should clarify whether they obtain $W=0$ for all the possible choices of ω_μ , ω_ν , S^1 or only for some specific choices.

Response: We thank the referee’s suggestions. The eigenvalue winding number is generally defined as $\mathcal{W} = \sum_{\mu \neq \nu} \left[\frac{1}{2\pi i} \oint_{S^1} (d\vec{\phi} \cdot \nabla_\phi \arg(\omega_\mu - \omega_\nu)) \right]$ with ω_μ, ω_ν denoting the complex eigenvalues of the Hamiltonian H and μ, ν indexing the states, namely $\mu, \nu = 1, 2, 3$ in our case. For how to choose one-sphere S^1 , as an explicit exemplification, we add a possible encircling loop S^1 in Fig. 2b, also as shown in Fig. R4.

The EL3s are embedded entirely in the ES2 and also osculate at the TPs. Besides, both the ES2 and EL3 run through the SBZ in the ϕ_1 -direction. In this case, any 2-sphere in the 3D parameter space must intersect with the ES2 and EL3, making the enclosure impossible. On the other hand, a closed loop, i.e., a 1-sphere S^1 (green dashed line) on the $\phi_2\phi_3$ -plane can still encircle both the ES2 and EL3, as shown in Fig. R4. Using the above equation, we obtain $\mathcal{W} = 0$. However, this is the net eigenvalue winding number carried by both the ES2 and EL3 because both are encircled. Therefore, the topology of the EL3 is still not characterized. This is the reason why we need to introduce the resultant winding numbers. We have added the closed encircling loop S^1 in the revised Fig. 2b and clarified it together with the eigenvalue winding number \mathcal{W} in the revised main text (in the first paragraph of the “Characterization of the EL3” section).

FIG. R4 The EL3 (red curves) and ES2 (blue surfaces) in the SBZ. The purple hexagons denote the TPs of EL3. The green dashed line denotes a S^1 loop encircling a TP.

(10) In the section “Characterization of the EL3”, the Authors introduce the resultant winding number in Eq. (2) that guarantees the stability of “EL3”. There, the Authors sometimes emphasize the role of symmetry. For example, the manuscript reads, “we introduce two types of symmetry-preserving perturbations” (Line 151) and “... the EL3 in Fig. 4a can be made locally stable without breaking the symmetry”. Here, I fail to clearly understand what symmetry they focus on and hence would like to request that the Authors specify which symmetry they consider.

Furthermore, if the Authors are to believe that symmetry protection is relevant to the stability of their “EL3”, such symmetry should be necessary for the introduction of the topological invariant in Eq. (2). However, the role of symmetry in Eq. (2) is unclear in the present manuscript; the Authors should clarify it.

In other words, if symmetry protection is indeed relevant to the stability of the Authors’ “EL3”, “EL3” should disappear immediately if symmetry-breaking perturbations are added. The Authors should explicitly confirm this by adding such symmetry-breaking perturbations.

Response: We thank the referee for these constructive suggestions. The PT-symmetry that our Hamiltonian [Eq. (1)] satisfies reduces the codimension from $2(n - 1)$ to $n - 1$, making the EP n locally stable in $(n - 1)$ -dimensional space. Algebraically speaking, the PT-symmetry makes the resultant $\mathcal{R}[p^{(j)}, p^{(j+1)}]$ become purely real, and $\Lambda = \eta + i\zeta = \mathcal{R}[p^{(1)}, p^{(2)}] + i\mathcal{I}[p, p^{(2)}]$ is then a 2-component real vector, implying Eq. (2) is adequate to digest the topology embedded in Λ . The topological stability of EL3 is guaranteed by the resultant winding number from Eq. (2). When two types of symmetry-preserving perturbations (δ_L and δ_M) are introduced, the PT-symmetry of the Hamiltonian remains, and thus the EL3 is locally stable and protected by the PT-symmetry, as shown in Fig. 4 in the main text. In contrast, when any symmetry-breaking perturbation is added, i.e., $L(\phi_1) = -60.86(1 - 0.50 \sin^2 \phi_1 + \delta'_L)$ with $\delta'_L = -0.1i$, the EL3 disappears since PT-symmetry is broken. There only exists the EL2 in the 3D parameter space, as shown in Fig. R5. To make the role of symmetry clear, we have added some discussions before Eq. (2) as

“Protected by the PT symmetry, the resultants η and ζ purely real, and the topology embedded in the Λ -field can be described by the winding numbers of Λ , defined as...”

Besides, we have added more information about the role of symmetry-breaking perturbations in Supplementary Note 5.

FIG. R5. EL2 in the 3D parameter space when PT-symmetry is broken. The blue (orange) surfaces correspond to $\text{Re}(\mathcal{D})=0$ [$\text{Im}(\mathcal{D})=0$]. The EL2 (blue solid lines) occurs as the intersections of the blue and red surfaces.

(11) On Line 154, while the Authors use “Bezout’s theorem”, they do not provide any explanations about it. The Authors should clarify what the statement of “Bezout’s theorem” is and why it is relevant here.

Response: We appreciate the referee’s suggestion. Bezout’s theorem refers to the number of intersection points of two curves in algebraic geometry. In particular, if two algebraic plane curves of degrees d_1 and d_2 have no component in common, they have $d_1 d_2$ intersection points, counted with their multiplicity, including points at infinity and points with complex coordinates².

In our manuscript, we determine the multiplicity of EP3s with Bezout’s theorem and then establish the link between the multiplicity and the resultants to comprehensively explain how to choose the proper resultants for topological characterization. According to the referee’s suggestions, more explicit explanations of Bezout’s theorem and a book on algebraic curves have been added to the revised main text [in the paragraph following Eq. (2)].

(12) On Line 177, the manuscript reads, “The EL3 demonstrated here together with previous works [34, 44, 48] show ...”. However, the relationship between this manuscript and these previous works [34, 44, 48] [i.e., K. Ding et al., Phys. Rev. X 6, 021007 (2016); W. Tang et al.,

Science 370, 1077 (2020); J. Hu et al. (2022)] is unclear solely from this sentence. Thus, I would like to request that the Authors clarify the similarities and differences with these previous works in more detail.

Response: We thank you for the kind suggestion. The comparison between the topological properties of higher-order EPs shown in the previous works and our manuscript is summarized in Table R2.

TABLE R2. Topological properties unique to higher-order EPs.

Higher-order EP structures	Unique topological properties	Main reference
Order-4 EP	Berry phase fractionally quantized to $\frac{3}{4}$.	Phys. Rev. X 6, 021007 (2016).
EP3 as a nexus for EP2 lines	Hybrid topological winding numbers and hybrid Berry phases.	Science 370, 1077 (2020).
EL3 as intersections of ES2	“Swallowtail catastrophe” structure in band dispersions.	Nat. Phys. (2023).
EL3 embedded in ES2	Eigenvalue topology diagnosable only in resultant manifolds, which is beyond the consideration of all the above cases.	This paper.

To make the relationship clear, we have enriched the relevant discussions in the revised main text (the last paragraph in the “Discussion and Conclusions” section) as

“The EL3 demonstrated here together with previous works show that the higher-order EPs possess far richer topological properties that are absent for both EP2 and Hermitian degeneracies. The hybrid topological winding number and the associated fractional Berry phases have been demonstrated to be a unique feature of higher-order EPs using the eigenvectors^{34,44}. Within the context of non-Hermitian bands, the higher-order EPs serve as the cusp singularities of multiple EL2s in the 3D space⁴⁴, and the topological characterization of EL2s viewing from the eigenvalue manifold necessitates the braid group^{50,51}, giving rise to the eigenvalue knots^{18,19,52} and non-Abelian conservation rule^{45,53}. Such the fact that the EL2s possess much more fruitful topological properties than the single EP2s also holds for the higher-order ELs, but the approach applied to the EL2s fails in the higher-order ELs. Our work here uncovers that EPs of different orders may form structures that challenge the conventional wisdom of topological characterization, and they necessitate an auxiliary resultant manifold, which remains well-behaved at the EP2 and only detects the EP3. Although the EL3 in this work are embedded in the ES2 originate from a single band gap, the resultant manifold approach can not only apply to the EL3 intersected by the ES2 from adjacent band gaps⁵⁴ but also be generalized to higher-order ELs⁴⁷, which paves the way to digest the topology of higher-order ELs in higher-dimensional non-Hermitian bands.”

Reply to Reviewer #2

In this work, authors report the realization of order-3 exceptional lines in coupled acoustic cavities.

However, I could not find the significance of this paper because a similar phenomenon is already reported in previous works.

Weiyuan Tang Science 370 6520 (2020).

Kun Ding, Phys. Rev. X 6, 021007 (2016).

Also, I could not find the novelty of the theoretical part because it is essentially the same as the discussion in Ref. [43] and Ref. [34].

For the above reason, I recommend the publication in another journal such as Scientific Reports.

Response: Thank you for the comments. The two papers mentioned by the referee are also our own works. We know very well that the present paper is substantially different from those ones, and the findings reported here represent significant advancements.

In *Kun Ding et al., Phys. Rev. X 6, 021007 (2016)*, higher-order EPs are proposed and observed for the first time. However, we were unaware of the conditions for EPs to form surfaces or lines. For this reason alone, the exotic topology of ES2 and EL3 could not possibly be conceivable.

In *Weiyuan Tang et al., Science 370 6520 (2020)*, a single EP3 was studied together with the exceptional arcs formed by EP2s. It was impossible for that EP3 to form EL3 in that system, let alone an EL3 embedded in ES2.

The main goal of this work is to study the topology of a higher-order EL, which is not the intersection of EPs of a lower order but is itself a submanifold of a topological manifold formed entirely by EPs. Such a situation has not been studied nor experimentally realized, and the topology of such an EL cannot be characterized by the prevailing methods based on homotopy groups but requires the consideration of the resultant manifolds. To further clarify the difference with previous works and highlight the key findings of this work, we have almost rewritten the “Discussion and Conclusions” section in the revised main text.

Reply to Reviewer #3

In the manuscript "Realization and Topological Properties of Third-Order Exceptional Lines Embedded in Exceptional Surfaces", the authors report both experimental and theoretical findings. Experimentally, they realize EP3 Line embedded in an EP2 surface within a three-dimensional periodic parameter space, utilizing coupled acoustic cavities. Theoretically, they propose the winding number of resultants to pinpoint the topological current of EP3 line embedded in an EP2 surface, since the winding number of eigenvalues is ill-defined in this situation.

The most remarkable aspect of their findings is the proposed resultant winding number, which enables the analysis of the topological charge and evolution of higher-order ELs when they are embedded in a lower-order EP manifold. I think this is an inspiring work, potentially offering significant contributions to the relevant research fields, especially PT-symmetric non-Hermitian systems. I can thus recommend it for publication in Nature Communications if the following points are clarified.

Response: We thank the referee for the complimentary summary and constructive comments. The followings are the responses and changes accordingly.

(1) The authors should clearly show the expression of discriminant \mathcal{D} at its first appearance in the manuscript. Readers may not be familiar with this formula.

Response: Following the referee's suggestion, we have shown the expression of discriminant \mathcal{D} when it first appears in the revised main text as

"... has entirely real discriminant $\mathcal{D} = \prod_{\mu < \nu} (\omega_{\mu} - \omega_{\nu})^2$ (where μ and ν are the eigenvalue indices), i.e., $\text{Im } \mathcal{D} = 0$ is always satisfied."

(2) The authors present the EP2 surface, EP3 line, and touching point in Fig.2(b) in an 8-fold larger synthetic Brillouin zone. The figure looks unclear and may hide the information that the touching point is only one point at the boundary of SBZ. I think all features could be clearly shown within a single SBZ. Alternatively, the authors should point out this information in the figure caption.

Response: The referee is correct. Indeed, the SBZ shown in Fig. 2(b) is not the first BZ. We choose to show the full SBZ because, experimentally, the parameters ϕ_1 , ϕ_2 , and ϕ_3 can freely run from $-\pi$ to $+\pi$. The trigonometric double-angle formula indeed reduces the terms in the Hamiltonian as $L(\phi_1) = \gamma + 2\kappa_1 \cos(2\phi_1)$, $M(\phi_2) = \epsilon + 2\kappa_2 \cos(2\phi_2)$, and $N(\phi_3) = \beta + 2\kappa_3 \cos(2\phi_3)$, and thus the SBZ in Fig. 2(b) can be regarded as the composition of 8 smaller *first* SBZs, with the touching point being only one point at the boundary, or SBZs with $\phi_1 \in [-\pi/2, +\pi/2]$,

$\phi_2, \phi_3 \in [-\pi, 0]$, and the touching point being at the center. By considering the parameter range experimentally, we think it is better to keep the full range of SBZ in Fig. 2(b) as it is but revise the relevant discussions in the second paragraph following Eq. (1) as

“We note that the SBZ here is an extended BZ consisting of eight identical copies of the first BZ. Physically, this is due to the quadratic dependence of the physical quantities (loss, hopping, and detuning) on the synthetic dimension; and mathematically, the trigonometric double-angle formula plays a role in Eq. (1). The choice of SBZ does not affect the validity of our analysis that follows.”

(3) In Fig. 3(a) and 3(c), the evolution of imaginary energy becomes crucial when the touching point is traversed in different parameter directions. The authors should show the corresponding evolution of imaginary energy when crossing the touching point in different directions.

Response: We thank the referee’s nice suggestion. We have displayed the measured imaginary parts of eigenfrequencies associated with the real parts in the revised Fig. 3b, d, f, as replicated in Fig. R6.

Fig. R6 Observation of EL3 and ES2. **a** The measured real and imaginary parts of eigenfrequencies along the dashed lines in Fig. 2a. An EP3 occurs when both the real and imaginary parts of three eigenfrequencies coalesce. **b-c** The measured real and imaginary parts of eigenfrequencies along the dashed lines in Fig. 2c and Fig. 2e are respectively shown in **b** and **c**. The circles in **a**, **b**, and **c** are experimental results. The red arrows point at the EP3s.

(4) Can the method of the resultant field be extended to higher-order EPn where $n > 3$? Is there a general principle for constructing the real and imaginary parts of the resultant vector field?

Response: We thank the referee for the insightful comment. The resultant field can be extended to higher-order EPs, and the recipe is depicted in Fig. R7. The key is that the resultants for the EPn topology must eliminate all the resultants already used in the lower-order EPs, say, the order less than n . Especially in our case, we shall avoid using $\mathcal{R}[p, p^{(1)}]$, which is nothing but the

discriminant (the red arrow), leading us to define the resultant winding number \mathcal{W}_Λ with $\eta = \mathcal{R}[p^{(1)}, p^{(2)}]$, $\zeta = \mathcal{R}[p, p^{(2)}]$ (the blue arrows). The exclusion of $\mathcal{R}[p, p^{(1)}]$ is because it contains singularities in itself (EP2s), and thus the connection between intersection multiplicity and resultant winding number fails. The generalization to the EP n is then straightforward, namely choosing $\mathcal{R}[p^{(j)}, p^{(n-1)}]$ with $0 \leq j < n - 1$. For example, the orange arrows in Fig. R7 depict the resultants used for the EP4. The topological description of the EP n then becomes the problem of characterizing the $(n - 1)$ -component complex vector. With additional symmetry imposed, such as the PT-symmetry in this work, the problem further reduces to characterizing the $(n - 1)$ -component real vector.

To clarify the recipe for the EP n , we have added three paragraphs, together with a new figure (Fig. 5), at the beginning of the revised ‘‘Discussion and Conclusions’’ section in the revised main text.

Fig. R7 Schematics for choosing resultants to characterize the EP n . The red, blue, and orange arrows denote the resultants chosen to digest the EP2, EP3, and EP4 topology, respectively. $p^{(j)}$ is the j th-order derivative of the characteristic polynomial with respect to ω .

(5) The authors describe the third-order exceptional line in Fig. 4(a) as ‘‘topologically neutral’’. Is there an additional symmetry contributing to the winding number of resultant field being zero? Could the authors provide further clarification on this ‘‘topologically neutral’’?

Response: We thank the referee for these excellent questions. The topological neutrality is not protected by the symmetry of the Hamiltonian. Instead, it is the consequence of the even dependence on ϕ_2 of the trigonometric functions. In Fig. 4c, the perturbation $\delta_M \neq 0$ in $M(\phi_2) = -38.61(\cos^2\phi_2 + \delta_M)$ does not break any symmetry, yet it breaks the topologically neutral EL3 into two pairs of EL3. When $\delta_M < 0$, i.e., the case shown in Fig. 4c in the main text, the pairs of oriented EL3 stably exist due to topological protection. When $\delta_M > 0$, the system no longer has EL3. So tuning δ_M from negative to positive induces the topologically oriented EL3 to merge and annihilate. It then becomes clear that at $\delta_M = 0$, the *topologically neutral* EL3 are a critical case. In a sense, this process is somewhat like the merging of two (Hermitian) Weyl points carrying opposite topological charges and the neutral case at $\delta_M = 0$ is like a charge-free Dirac point. To further clarify the topological neutrality, we have added the following discussions to the last paragraph of the revised ‘‘Characterization of the EL3’’ section as

“Indeed, the currents cancel when the two pairs of EL3 merge at $\delta_M = 0$. When δ_M is increased to positive, the EL3 vanish from our system. In other words, the topological currents defined by the winding of Λ are able to capture to merging and annihilation of the EL3.”

(6) There are a few typographical errors: In the caption of Fig. 4: "the field (arrows) in of"; On page 8 of Supplemental Information: "we define two different resultant fields ...".

Response: Thank you for the careful reading. The typos have been corrected in the revised version.

References

1. Garrett, S. L. *Understanding Acoustics: An Experimentalist's View of Acoustics and Vibration*. (Springer, 2017).
2. Walker, R. J. *Algebraic curves*. (Springer-Verlag, 1978).

REVIEWERS' COMMENTS

Reviewer #1 (Remarks to the Author):

I would appreciate the response and the corresponding revision of the manuscript. The Authors have addressed most of my concerns satisfactorily and revised the manuscript accordingly. Now, I would like to recommend publication of this manuscript in Nature Communications.

Reviewer #2 (Remarks to the Author):

In the revised manuscript authors explained the differences from the following papers
Weiyuan Tang Science 370 6520 (2020).
Kun Ding, Phys. Rev. X 6, 021007 (2016).

However, they are just technical differences which is not sufficient to deserve the Nat. Comm. In addition, the topology of EP3 EP4, EL3, and their variants has already been elucidated in Refs. [38] and [39] (i.e., Delplace et al., and Mandal et al.,), which means that elucidating their topology is not a newly obtained result.

Therefore, I recommend the publication in another journal such as Scientific Reports.

Reviewer #3 (Remarks to the Author):

I think the authors have addressed all my questions and concerns in this revised version. Given the theoretical novelties, particularly the topological characterization of higher-order EPs embedded in a lower-order EPs manifold, as well as the relevant experimental realizations, I now recommend the manuscript for publication in Nature Communications.

Reply to Reviewer #1

I would appreciate the response and the corresponding revision of the manuscript. The Authors have addressed most of my concerns satisfactorily and revised the manuscript accordingly. Now, I would like to recommend publication of this manuscript in Nature Communications.

Response: We sincerely thank the referee for recommending the publication of our work. All the suggestions and comments are appreciated.

Reply to Reviewer #2

*In the revised manuscript authors explained the differences from the following papers
Weiyuan Tang Science 370 6520 (2020).
Kun Ding, Phys. Rev. X 6, 021007 (2016).*

However, they are just technical differences which is not sufficient to deserve the Nat. Comm. In addition, the topology of EP3 EP4, EL3, and their variants has already been elucidated in Refs. [38] and [39] (i.e., Delplace et al., and Mandal et al.,), which means that elucidating their topology is not a newly obtained result.

Therefore, I recommend the publication in another journal such as Scientific Reports.

Response: We thank the reviewer for raising the two concerns. First, this work does not differ solely from the technical side with our two previous publications [Science 370 6520 (2020) and Phys. Rev. X 6, 021007 (2016)], but introduces an approach based on the resultant manifold to digest the topology carried by the EL3 embedded in the ES2, which has not been revealed before. This leads to the second concern. References [38] and [39] are two significant works that reveal the symmetry and resultant criteria required to achieve higher-order EPs. However, they do not aim to digest the topology of higher-order exceptional lines embedded in the manifold formed by the lower-order EPs. Thus, their theory cannot be applied to describe and predict the possible local evolutions of EL3s investigated in our work. To be precise, we do wish to emphasize several essential developments in the following:

- The critical difference between our work and Ref. [38] is that we disclose a unique choice of the resultants, which has been proved based on the relation between intersection multiplicity and resultant winding number, to digest the topology of EL3 embedded in the ES2. Such an approach ignores the influence of EP2, only detects EP3, and is successful in topologically

characterizing the EP3s despite the presence of the EP2s. In short, as depicted in Fig. R1, the choice of resultants for EP n must eliminate all the resultants already used in the lower-order EPs, say, the order less than n . Especially in our case, we must avoid using $\mathcal{R}[p, p^{(1)}]$, which is nothing but the discriminant in our three-level system, leading us to define the resultant winding number \mathcal{W}_Λ with $\Lambda = \eta + i\zeta$ and $\eta = \mathcal{R}[p^{(1)}, p^{(2)}], \zeta = \mathcal{R}[p, p^{(2)}]$. The choice of resultants in Ref. [38] is $\mathcal{R}[p^{(j)}, p^{(j+1)}]$ with $0 \leq j < n - 1$, which can identify the locations of EPs but is not suitable to evaluate their topological properties.

- Our approach can be generalized to higher-order EPs, and the recipe is depicted in Fig. R1. The generalization to the EP n is to choose $\mathcal{R}[p^{(j)}, p^{(n-1)}]$ with $0 \leq j < n - 1$. For example, the orange arrows in Fig. R1 depict the resultants used for the EP4. The topological description of the EP n then becomes the problem of characterizing the $(n - 1)$ -component complex vector. With additional symmetry imposed, such as the PT-symmetry in this work, the problem further reduces to characterizing the $(n - 1)$ -component real vector.

FIG. R1. Schematics for choosing resultants to characterize the EP2, EP3, EP4, and EP n . The red, blue, and orange arrows denote the resultants chosen to digest the EP2, EP3, and EP4 topology, respectively. $p^{(j)}$ is the j th-order derivative of the characteristic polynomial with respect to ω .

- Last but not least, our work here is the first experimental demonstration of not only an EL3 but also non-Hermitian singularities that have to be characterized using resultant winding numbers. In Ref. [38], the EP3 and EL3 are theoretically investigated using the linearized rotating shallow water model. It is a model that is very interesting in its own right, and it is no doubt an impressive extension of non-Hermitian models to potentially an entirely new realm of physics. However, acoustic platforms are more versatile by offering immense and more detailed controls of system parameters (thanks to the years of development in phononic crystals, metamaterials, and non-Hermitian acoustics), which are essential for realizing sophisticated models in this work.

To fully address these concerns, we have already substantially revised the manuscript (the “Characterization of the EL3” section) to strengthen the relevant discussions and highlight the role of proper resultants. In particular, three paragraphs (in the “Discussion” section) and a new figure (Fig. 5) have been added to the revised main text. We believe these revisions already can differentiate our work from Ref. [38]. (Ref. [39] did not mention the topological characterization of EP3s.)

Reply to Reviewer #3

I think the authors have addressed all my questions and concerns in this revised version. Given the theoretical novelties, particularly the topological characterization of higher-order EPs embedded in a lower-order EPs manifold, as well as the relevant experimental realizations, I now recommend the manuscript for publication in Nature Communications.

Response: We appreciate the reviewer’s efforts in evaluating our work and thank for the recommendation of publication. We are grateful for the constructive comments/suggestions.